# Towards Detecting Contextual Real-Time Toxicity for In-Game Chat

**Zachary Yang**
Ubisoft La Forge
McGill University, Mila
zachary.yang@mail.mcgill.ca

**Nicolas Grenon-Godbout**
Ubisoft UDO
nicolas.grenon-godbout@ubisoft.com

**Reihaneh Rabbany**
McGill University, Mila
CIFAR AI chair
reihaneh.rabbany@mila.quebec

## Abstract

Real-time toxicity detection in online environments poses a significant challenge, due to the increasing prevalence of social media and gaming platforms. We introduce ToxBuster, a simple and scalable model that reliably detects toxic content in real-time for a line of chat by including chat history and metadata. ToxBuster consistently outperforms conventional toxicity models across popular multiplayer games, including Rainbow Six Siege, For Honor, and DOTA 2. We conduct an ablation study to assess the importance of each model component and explore ToxBuster's transferability across the datasets. Furthermore, we showcase ToxBuster's efficacy in post-game moderation, successfully flagging 82.1% of chat-reported players at a precision level of 90.0%. Additionally, we show how an additional 6% of unreported toxic players can be proactively moderated.

## 1 Introduction

Online spaces are plagued by toxic speech, spanning social media platforms (e.g., Facebook (Ciftci et al., 2017), Twitter (Watanabe et al., 2018), Reddit (Mohan et al., 2017), YouTube (Döring and Mohseni, 2020)), in-game chats (Silva et al., 2020) and news websites comment sections (Zannettou et al., 2020). As evidenced by surveys from the Anti-Defamation League (ADL), exposure to toxic language not only alienates users but also poses a range of psychological harms and the potential to incite real-world violence (ADL, 2021). Even worse, marginalized groups continue to face a disproportionate level of targeted online hate and harassment.

Companies strive to foster a healthy online community and have employed various methods to address toxic speech, such as censoring words, (shadow) banning users or blocking them from communicating (Maher, 2016; Lewington, 2021). However, the vast amount of user-generated data and the rapidly evolving nature of language have made it exceedingly challenging to implement consistent moderation practices.

We leverage recent advancements in large language models to create accurate and transferable models for effective content moderation. Contextual language embeddings, like BERT (Devlin et al., 2018), serve as the foundation for many state-of-the-art toxic speech detection models. However, most existing approaches either neglect context entirely or yield only marginal improvements and are not fit for real-time moderation for in-game chat.

To address this limitation, we propose ToxBuster, the first real-time in-game chat toxicity detection model capable of integrating *chat history* and *metadata*. It is trained on annotated datasets that encompass diverse perspectives, with annotators from marginalized groups offering valuable insights. Furthermore, we demonstrate the transferabilty of ToxBuster across different games, as well as its adaptability to a completely different domain, i.e., comment threads of news articles. When it comes to post-game moderation, we prioritize high precision settings to maximize the impact of our model without overloading limited manual content moderation resources with false positives. Remarkably, ToxBuster achieves an impressive 82% identification rate for toxic players—reported by other players for posting toxic chat— at a precision level of 90.0%. This can significantly reduce the moderation load of human moderators. In summary, contributions are threefold:

- We present ToxBuster, a real-time toxicity detection model for in-game chat that outperforms current available solutions by leveraging chat history and metadata
- We show that the proposed model can transfer across different games and domains.
- We discuss the post-game moderation implications and show how ToxBuster can automate a high percentage of obvious cases of toxicity and

refocus moderator's effort towards unclear cases.

**Reproducibilty:** Source code for ToxBuster is hosted on Github. Due to legal and privacy reasons, R6S and FH datasets cannot be public.

## 2 Related Works

Toxicity detection research has gained increasing attention due to the challenges it poses. These challenges include not only the text itself, such as the presence of out-of-vocabulary words, but also the lack of consensus on the precise definition of toxicity (van Aken et al., 2018). Definitions vary, ranging from a broad concept of toxicity (Georgakopoulos et al., 2018) to more specific categories like hate speech (Gambäck and Sikdar, 2017), abusive language (Park and Fung, 2017), cyberbullying (Zhong et al., 2016), and offensive language. In our study, we focus on adapting categories defined by Alliance (2020) to address disruptive behavior in online games.

Traditionally, toxicity detection has been approached as a classification task using various methods, including traditional ML models with manual feature engineering (Watanabe et al., 2018), deep neural networks (Gambäck and Sikdar, 2017; Zhong et al., 2016), and pretrained language models (Almerekhi et al., 2022; Jhaveri et al., 2022; Lees et al., 2022). Previous studies have explored including additional context such as news article titles (Gao and Huang, 2017), usernames (Mubarak et al., 2017), Twitter user metadata (Fehn Unsvåg and Gambäck, 2018), game-specific slang (Weld et al., 2021) and parent comments and discussion titles (Pavlopoulos et al., 2020). These attempts to incorporate additional context into language models have yielded limited performance gains, typically less than 1%. To address this, we incorporate more than just the previous comment (Pavlopoulos et al., 2020; Yu et al., 2022) by utilizing all preceding chat history available as well.

While previous research on toxicity detection has explored metadata aspects, none have addressed the conversational aspect, especially with this level of granularity, in contexts where more than just the preceding line of text is used. We draw inspiration from multi-turn conversational models and incorporate speaker segmentation (metadata including TeamID, Chat Type, and PlayerID) and a naive dialogue augmentation technique that has shown to enhance BERT for multi-turn conversations (Lu et al., 2020).

## 3 Methodology

In this section, we present four toxicity datasets along with ToxBuster and the baselines models.

### 3.1 Dataset

We collected and curated datasets from three multiplayer games representing different genres: Rainbow Six Siege (R6S), For Honor (FH), and Defense of the Ancients 2 (DOTA 2). R6S is a first-person shooter, FH is a melee action game, and DOTA 2 is an online battle arena game. Additionally, we incorporated Jigsaw's Civil Comments (CC) dataset into our research. The annotation process involved token-level annotations for R6S and FH, while DOTA 2 and CC were annotated at the sentence-level. For games, chat lines from a match is considered a document while for CC, a document contains each comment on an article. For detailed dataset statistics, please refer to Table 1.

|  | R6S | FH | DOTA 2 | CC |
|---|---|---|---|---|
| Domain | Game | Game | Game | Media |
| Time | 2021 | 2022 | 2017 | 2018 |
| No. of Documents | 1,392 | 5,340 | 1,921 | 65,148 |
| No. of Lines | 95,612 | 99,371 | 62,483 | 131,319 |
| Avg. WPL | 3.20 | 4.01 | 2.41 | 54.63 |
| Avg. LPD | 69.07 | 19.46 | 32.53 | 2.01 |
| % of Toxic Lines | 32.06% | 21.24% | 23.91% | 8.01% |

Table 1: Toxic chat datasets overview. WPL - Words per Line. LPD - Lines per Document.

### 3.1.1 R6S & FH

For each game, we extract chat logs from regions that communicate prominently in English. We oversample matches with a high volume of chat lines and/or instances where at least one player was reported by another player. The toxic classes we adopt align with the "Disruptive Behavior in Online Game" categories outlined by Alliance (2020): **Hate and Harassment**, **Threats**, **Minor Endangerment**, **Extermism**, **Scams and Ads**, **Insults and Flaming**, **Spam** and **Other Offensive Text**. For detailed definitions of each class and annotation guidelines, please refer to Appendix A and B respectively.

Our dataset includes diverse perspectives by including annotators from marginalized groups. Each line of the dataset was reviewed by three annotators. To aggregate labels, we used the minimum intersecting span of words from the annotators and employed a majority vote, resolving ties by assigning the most severe class. The final number of chat

lines per class and Fleiss $\kappa$ score are reported in Table 2.

| Class | R6S | FH |
|---|---|---|
| Hate and Harassment | 5,482 | 4,453 |
| Threats | 618 | 421 |
| Minor Endangerment | 625 | 109 |
| Extremism | 392 | 173 |
| Scams and Ads | 456 | 53 |
| Insults and Flaming | 8,824 | 11,329 |
| Spam | 11,127 | 2,210 |
| Other Offensive | 3,117 | 2,077 |
| Non-toxic | 64,937 | 78,292 |
| Fleiss $\kappa$ | 0.54 | 0.47 |

Table 2: Class distribution for R6S and FH (ordered by toxicity severity level, descending)

### 3.1.2 DOTA 2

We utilize Weld et al. (2021)'s dataset as the foundation. This dataset comprises automatic token-level annotations and four manual sentence-level annotations: "explicitly-toxic", "implicitly-toxic", "action", and "other". To adapt the dataset for real-time toxicity detection, we introduce the following modifications: 1) merging the "action" class with the "other" class, 2) breaking down the merged sentences back into lines of chat, addressing the merging of consecutive lines originating from the same user. When a merged sentence is "explicitly-toxic", chat lines are labeled as "explicitly-toxic" if they contain at least one toxic token. Otherwise, they are labeled as "other". For all other classes, the chat lines retain the original labels from the merged sentence.

| Class | DOTA 2 |
|---|---|
| Explicitly Toxic | 10,511 |
| Implicitly Toxic | 4,431 |
| Other | 47,541 |

Table 3: Class distribution for DOTA 2

### 3.1.3 CC

Civil Comments platform is a commenting plugin for independent news sites. To adapt Jigsaw's Civil Comments dataset for real-time toxicity detection, we create its comment history by including the past comment thread. This is achieved by tracing all parent comments. We use the binary toxic versus non-toxic label for each comment.

## 3.2 ToxBuster

ToxBuster is a token classification model based on $BERT_{BASE}$. In our experiments, we employed a 60-20-20 train-validation-test split using 5 different random seeds. We present the mean and standard deviations of each metric. Our model achieved improved performance through two approaches: **Chat History** and **Chat Speaker Segmentation**.

### 3.2.1 Chat History

To enable reliable real-time toxicity detection, we utilize chat history as additional context. For all three in-game chat datasets, the average Word Per Line (WPL) is less than 5. Thus, including all available chat history becomes crucial.

Drawing inspiration from the question-answering task, we treat chat history as the *question* and the current chat line as the *answer*. In particular, we concatenate all chat lines preceding the current chat line as chat history and separate them with the "[SEP]" token as the *question*. During training, only the toxic class label is provided for the current chat line, i.e. our model is trained to predict the probability of each token's class in the current chat line given the chat history.

$BERT_{BASE}$ has a max token size of 512. Since the main goal is to predict the current token's class, we employ a custom truncator. It first prioritizes truncating the chat history by removing the earliest chat lines (truncating left) and, if necessary, the current chat line on the right. For the three in-game chat datasets, the current chat line would never be truncated.

In the case of in-game chat, players have the option to communicate with their teammates or with everyone. Following the approach of dialogue augmentation (Lu et al., 2020), we determine the type of previous chat lines to include in the chat history by considering four different scopes: *personal*, *team*, *global* and *moderator*. The *personal* scope of the chat history includes only lines written by the current player. The *team* scope expands the chat history to include lines from players on the same team. The *global* scope further includes past chat lines broadcasted to all players. The *moderator* scope includes remaining chat lines, such as those from enemy teams that are not broadcasted to all players. Table 4 presents a fabricated sample chat at the beginning of a match.

| # | Line | *playerID* | *chatType* | *teamID* |
|---|------|-----------|-----------|---------|
| 1 | (Team) Apple: Hf | 6 | Team | 1 |
| 2 | (All) Banana: Hf | 5 | All | 1 |
| 3 | (Team) Grape: Which site? | 1 | Team | 0 |
| 4 | (Team) Orange: A | 0 | Team | 0 |
| 5 | (All) Orange: Glhf | 0 | All | 0 |

Table 4: Chat history & scope. We present the metadata encoding when considering line #5. The corresponding chat history for the different scopes of *personal*, *team*, *global* and *moderator* would be then line 4, lines 3-4, lines 2-4, and lines 1-4.

### 3.2.2 Chat Speaker Segmentation

The primary objective is to enhance the model by incorporating conversational and game-related information. To achieve this, we introduce speaker segmentation, which includes three metadata attributes for each chat line: *playerID*, *chatType*, and *teamID*.

ToxBuster is a BERT-based model with an additional three inputs provided. We follow BERT's input embedding scheme, where it is the sum of all (position and token) its encoding. As shown in Figure 1, we introduce an encoding for each of the *teamID*, *chatType* and *playerID* that corresponds to the token. As such, the new input embedding would then be the sum of all 5 encodings: Token, Position, *teamID*, *chatType* and *playerID*.

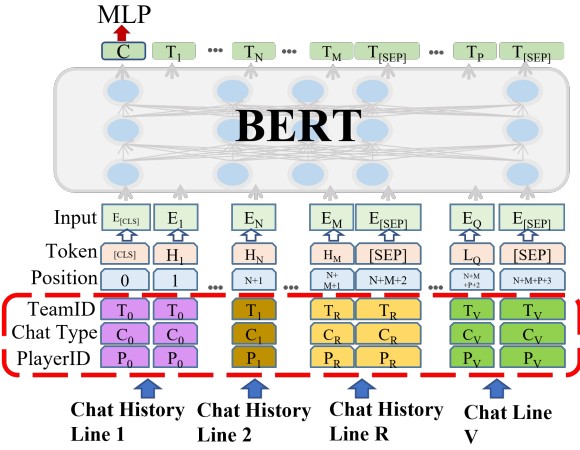

Figure 1: ToxBuster with chat speaker segmentation. Input embeddings are the sum of the corresponding token, position, teamID, chat type and playerID. The chat history includes as many lines as possible.

For the three chat metadata, *playerID* and *teamID* dynamically changes based on the player of the current chat line. *ChatType* can be either **team** or **all**, indicating whether a line is exclusive to the **team** or broadcasted to **all** players. *PlayerID*

is the unique identifier for the player associated with the chat line, starting from **0** and bounded by the number of teams times the team size. For consistency, the player of the current chat line is always **0**. For other players on the same team, the identifier is incremented based on the recency of that player's chat line. For players on the other team, the identifier starts from the size of the team, e.g, in a 5 v 5 game, the most recent opponent that has typed in chat will be **5**. With this scheme, the *playerID* can be extended to even Battle Royal games, where there can be multiple enemy teams. *TeamID* is the unique identifier of the team the current player belongs to. For consistency, the current player is always team **0**. The enemy team would be team **1**. For battle royal games, this scheme can be extended similarly to the *playerID*. The last three columns in Table 4 describe the *playerID*, *chatType* and *teamID* when detecting toxicity for line 5.

### 3.3 Baselines

Here, we present three baseline models used for comparison with ToxBuster.

#### 3.3.1 Cleanspeak

Cleanspeak is a paid tool that has "premier profanity filter and moderation"[1] based on user-defined keywords and regexes. Toxicity is determined based on the API response containing matched text related to various toxic classes, which can be mapped to our own. Currently, the toxic classes are "bigotry_racism", "harm_abuse", "threats", "grooming", "terrorism", "pii" (personal identifiable information), "spam", "bullying", "vulgarity", "sexual", "alcohol_drug", "asciiart".

#### 3.3.2 Perspective API

We utilized Perspective API[2] (*v1alpha1*) developed by Jigsaw and Google's Counter Abuse Technology team for promoting healthier online discussions (Lees et al., 2022). As noted by the research team, we classified a chat line as toxic only if the returned toxic score is >= 0.7. We remove approximately 13% of the chat lines during the calculation of the metrics due to the API returning an error code for unsupported languages.

#### 3.3.3 Detoxify

Detoxify$_{unbiased}$ (Hanu and Unitary team, 2020) is a BERT-based models trained on CC. Further

---

[1] https://cleanspeak.com/docs/3.x/tech/apis/
[2] https://perspectiveapi.com/

details of how the models were trained can be found in their repository [3].

# 4 Results and Discussions

In the upcoming sections, we analyze ToxBuster's performance relative to baselines, examine individual toxic class performance, conduct an ablation study to evaluate each model component's impact and assess implications for post-game moderation.

## 4.1 Baseline Comparison

Table 5 compares ToxBuster with baseline models across all 4 datasets. The baseline models focus on the binary task of classifying chat lines as toxic or non-toxic. ToxBuster is trained on each dataset's specific toxic class, while ToxBuster$_{base}$ lacks chat history and speaker segmentation.

ToxBuster$_{full}$ achieves the highest performance, with a 5% improvement on R6S and a 3% improvement on FH compared to ToxBuster$_{base}$. However, no significant improvement is observed for DOTA 2 and CC, possibly due to dataset characteristics. DOTA 2 dataset includes only **all** chat and initial annotation included merged chat lines. Similarly, the CC dataset lacks consideration for contextual information during annotation.

In terms of precision, Perspective API generally outperforms Cleanspeak, while Detoxify$_{unbiased}$ excells in recall and F1 score. Cleanspeak ranks the lowest. Notably, baseline models perform poorly on the DOTA 2 dataset, likely due to game slang and a low average WPL of 2.41.

## 4.2 ToxBuster Transferabilty

In this experiment, ToxBuster is trained using one specific toxic dataset and evaluated on the remaining datasets. Each dataset was transformed into a binary classification task, determining whether each chat line was toxic or non-toxic. The findings are summarized in Table 6, which reveals interesting patterns.

*Correlation with WPL:* When considering the diagonal entries (trained and tested on the same dataset), we observe a positive correlation between performance and the average Word Per Line (WPL). The average WPL values for DOTA 2, R6S, FH and CC are 2.41, 3.20, 4.01, and 54.63, respectively. This matches our intuition where greater word count facilitates more reliable toxicity prediction.

---

[3] https://github.com/unitaryai/detoxify

*Correlation with Domain:* Focusing on the top left section, the performance between R6S and FH was consistently good, ranging from 81 to 89. Expanding to include DOTA 2, we observe an acceptable performance of 72, but it drops to 59 when CC is included. This suggests that R6S and FH share similar toxic language characteristics, with some similarities extending to DOTA 2 and fewer with CC. We attribute this pattern to the fact that R6S, FH, and DOTA 2 belong to the game chat domain, while CC falls within the social media comment threads domain.

*Correlation with Time:* If we shift our attention to the bottom right section, the performance between DOTA 2 and CC also exhibit good results, ranging from 81 to 95. Interestingly, DOTA 2 performed better on CC than either R6S or FH. We attribute this difference to the evolution of toxic language over time. The CC and DOTA 2 datasets date back to 2017 and 2018, respectively, while the R6S and FH datasets are more recent, from 2021 and 2022.

In summary, for robust real-time toxicity detection in game chat, the model should be trained on up-to-date toxic game chat datasets rather than generic toxicity datasets.

## 4.3 Game Adaptation

We also assess adapting ToxBuster from R6S to FH, two multiplayer games with slightly different data, including game-specific references and potential differences in toxicity. We use 20,339 chat lines (20% of the dataset) as the test set. To compare fine-tuning and adapting, we fine-tune ToxBuster on FH and perform transfer learning on the best-performing ToxBuster model on R6S with gradually increasing the size of the FH training dataset. The results are presented in Table 7. As expected, transfer learning outperforms fine-tuning in all scenarios, indicating good transferability between R6S and FH. The performance differences amongst the transfer learning settings are minimal, suggesting that at low data settings, transfer learning may not provide significant improvements and could even slightly decrease performance (+691 indicates lower overall performance). However, adapting from R6S and FH is beneficial as it demonstrates a boost in the model, as evident from the 1.5% in precision from ToxBuster$_{+62,528}$ and ToxBuster$_{R6S+62,528}$.

| | R6S | | | FH | | | DOTA 2 | | | CC | | |
|---|---|---|---|---|---|---|---|---|---|---|---|---|
| | P | R | F1 | P | R | F1 | P | R | F1 | P | R | F1 |
| Cleanspeak | 66.62 | 29.10 | 40.48 | 65.91 | 38.92 | 48.93 | 29.47 | 3.07 | 5.55 | 25.14 | 67.56 | 36.65 |
| Perspective API | 75.11 | 24.38 | 36.81 | 73.48 | 37.97 | 50.07 | 11.47 | 1.06 | 1.95 | 83.63 | 60.98 | 70.53 |
| Detoxify$_{unbiased}$ | 63.47 | 29.58 | 40.33 | 66.01 | 50.09 | 56.98 | 26.98 | 3.22 | 5.75 | 69.99 | 75.81 | 72.79 |
| ToxBuster$_{base}$ | 77.21 | 77.91 | 77.36 | 80.41 | 82.17 | 81.88 | 83.98 | 84.82 | 84.05 | 94.53 | 94.43 | 94.45 |
| ToxBuster$_{full}$ | **82.95** | **83.56** | **83.25** | **84.88** | **85.62** | **85.09** | **84.39** | **85.09** | **84.53** | **95.48** | **95.46** | **95.47** |

Table 5: Toxicity classification performance across datasets. Performance is measured in terms of weighted average precision, recall, and F1 score. Baseline models focus on the binary task of toxic versus non-toxic, while ToxBuster is trained and tested for each toxic class.

| | | Test | | | |
|---|---|---|---|---|---|
| | | R6S | FH | DOTA 2 | CC |
| Train | R6S | 85.38 | **88.65** | **77.70** | 59.88 |
| | FH | **81.79** | 89.60 | **81.62** | 42.27 |
| | DOTA 2 | **72.81** | **85.54** | 85.14 | **89.75** |
| | CC | 67.49 | 75.77 | **81.88** | 95.47 |

Table 6: ToxBuster transferabilty across datasets. Transferabilty is tied closely with domain & time.

| | ToxBuster | | ToxBuster$_{R6S}$ | |
|---|---|---|---|---|
| | Precision | Recall | Precision | Recall |
| 0 | - | - | 84.14 ± 0.3 | 85.81 ± 0.3 |
| + 691 | 76.50 ± 1.2 | 81.38 ± 0.4 | 84.01 ± 0.4 | 85.18 ± 0.4 |
| + 15,787 | 83.75 ± 0.6 | 84.34 ± 0.5 | 84.57 ± 0.1 | 84.27 ± 0.0 |
| + 25,448 | 84.20 ± 0.3 | 85.24 ± 0.5 | 85.04 ± 0.3 | 85.43 ± 0.3 |
| + 35,410 | 84.47 ± 0.4 | 85.31 ± 0.4 | 85.11 ± 0.3 | 85.65 ± 0.6 |
| + 53,582 | 84.84 ± 0.8 | 85.47 ± 0.9 | **85.43 ± 0.5** | 85.89 ± 0.3 |
| + 62,528 | 84.88 ± 0.6 | 85.62 ± 0.6 | 85.23 ± 0.4 | **85.93 ± 0.6** |

Table 7: Comparison of fine-tuning ToxBuster and adapting ToxBuster$_{R6S}$ with increasing number of training data from FH. ToxBuster$_{R6S}$ achieves equal performance as ToxBuster with less than half the training data.

| | R6S | | FH | |
|---|---|---|---|---|
| Class | Precision | Recall | Precision | Recall |
| Hate & Harass | 63.78 ± 2.3 | 56.40 ± 3.8 | 58.76 ± 5.8 | 45.55 ± 0.4 |
| Threats | **31.53 ± 3.7** | **22.85 ± 4.6** | 36.38 ± 8.6 | 27.36 ± 7.0 |
| Minor Endanger. | 38.28 ± 7.1 | 29.21 ± 3.7 | **27.37 ± 13.0** | **13.70 ± 7.8** |
| Extremism | 54.58 ± 8.0 | 40.86 ± 8.9 | 35.08 ± 13.2 | 20.38 ± 10.6 |
| Scams & Ads | 56.89 ± 5.0 | 45.62 ± 9.5 | 51.33 ± 27.2 | 20.16 ± 10.1 |
| Insults | 58.97 ± 3.5 | 53.72 ± 2.3 | **59.63 ± 0.9** | **64.11 ± 2.3** |
| Spam | **84.15 ± 3.8** | **78.42 ± 3.8** | 58.86 ± 4.6 | 46.51 ± 9.1 |
| Other Offensive | 47.52 ± 4.2 | 44.20 ± 3.0 | 33.33 ± 4.1 | 20.23 ± 3.6 |
| Non-toxic | 88.32 ± 0.8 | 91.85 ± 0.9 | 92.55 ± 0.4 | 94.39 ± 0.4 |

Table 8: ToxBuster class performance for R6S and FH. Blue and red highlights the best and worst performing toxic classes respectively.

reveals that 25% of them were clearly non-toxic. However, 42% of the text contained sexual content unrelated to the game, which could be categorized as bordering on sexual harassment. Additionally, 21% of the text mentioned racial groups, where the toxicity of the chat line was open to debate. Furthermore, 12% of the text included racial terms that implied toxicity based on the context. For false negatives, our analysis indicates that 65% of them were toxic but not caught due to misspellings. Furthermore, 25% of the false negatives contained implicit racism, ageism, and sexism. Additionally, 5% of the false negatives were considered implicitly toxic based on the context. Lastly, 5% were annotated as toxic due to the use of profanity, although they were not used in a toxic manner. These findings highlight the nuanced nature of evaluating toxicity.

## 4.6 Ablation Study

In this section, we conduct an ablation study on the two main components: **Chat History** and **Chat Speaker Segmentation**

### 4.6.1 Chat History

In this experiment, we evaluate the impact of chat history without chat speaker segmentation, as shown in Table 9. Intuitively, increasing the amount of context improves the reliability of toxicity prediction. This is evidenced by the nearly

## 4.4 Class-wise Evaluation

We also analyze our model's performance for each toxic class with results shown in Table 8. The model can easily differentiate amongst non-toxic, toxic words and spam. We attribute the lower F1 score in threats and minor endangerment to their minority. While extremism and scams and ads have even fewer samples, the language for both these two categories are usually very unique. We notice that the model often confuses amongst hate and harassment, threats, other offensive as the words are often very similar and additional context from chat history, in game knowledge and social constructs are needed. Annotators also reported it was often hard to choose between these categories as well.

## 4.5 Error Analysis

We randomly sampled 100 false positives and 100 false negatives. For false positives, our analysis

4% performance improvement with either *global* or *moderator* scope. Notably, precision increases as we include more context, while *global* demonstrates higher recall and F1 score compared to *moderator*, albeit with slightly lower precision. These findings align with Lu et al. (2020) who emphasized the importance of dialogue consistency during classification.

|  | Precision | Recall | F1 Score |
|---|---|---|---|
| No History | 77.21 ± 1.3 | 77.91 ± 1.0 | 77.36 ± 1.3 |
| Personal | 79.11 ± 0.7 | 79.82 ± 0.3 | 78.89 ± 0.8 |
| Team | 80.51 ± 0.2 | 80.10 ± 0.2 | 81.30 ± 0.0 |
| Global | 81.60 ± 0.5 | **82.21 ± 0.4** | **81.70 ± 0.5** |
| Moderator | **81.90 ± 0.4** | 81.47 ± 0.4 | 81.68 ± 0.4 |

Table 9: Impact of chat history & scope. Performance increases with larger scopes of chat history.

### 4.6.2 Chat Speaker Segmentation

In this experiment, we focus on the impact of chat speaker segmentation while keeping the chat history constant in the *global* mode. We compare two methods of including chat metadata: in-line and chat speaker segmentation. The in-line method appends the metadata in front of each chat line, which intuitively incorporates the information. Table 10 presents the specific impact of each metadata added and the overall impact when all three are included. These metadata are typically considered during human annotation to assess chat consistency. Across both methods, *playerID* is the most influential feature, as it distinguishes chat lines based on the player. The *chat type*, indicating whether the intended audience is the team or everyone (including the enemy team), is the next significant feature. Comparing in-line and chat speaker segmentation, the latter outperforms in terms of precision, recall, and F1 score for each metadata. We speculate that ToxBuster can better learn the correlation between chat line metadata and toxicity when each token has its own associated metadata, rather than positioning it in front of the chat line.

### 4.7 Post-Game Moderation Implication

In a real-world scenario, ToxBuster would be operated at a high precision threshold to ensure accurate detection of toxic chat and minimize false positives. As shown before in Table 5, ToxBuster$_{full}$ significantly outperforms Cleanspeak and Perspective API, achieving 82.95% (+7) in precision and 83.56% (+54) in recall.

|  |  | Precision | Recall | F1 Score |
|---|---|---|---|---|
|  | Base | 81.60 ± 0.5 | 82.21 ± 0.4 | 81.70 ± 0.5 |
| In-Line | w/ teamID* | 81.59 ± 0.5 | 82.08 ± 0.9 | 81.69 ± 0.7 |
|  | w/ chat type* | 81.82 ± 0.5 | 82.38 ± 0.4 | 81.88 ± 0.4 |
|  | w/ playerID* | 81.90 ± 0.4 | 82.47 ± 0.4 | 82.01 ± 0.4 |
|  | w/ *full** | **81.95 ± 0.3** | **82.56 ± 0.3** | **82.12 ± 0.3** |
| Speaker S. | w/ teamID | 81.61 ± 0.4 | 82.23 ± 0.4 | 81.72 ± 0.4 |
|  | w/ chat type | 81.91 ± 0.5 | 82.53 ± 0.8 | 82.18 ± 0.6 |
|  | w/ playerID | 82.36 ± 0.4 | 82.79 ± 0.4 | 82.43 ± 0.4 |
|  | w/ *full* | **82.95 ± 0.3** | **83.56 ± 0.3** | **83.25 ± 0.3** |

Table 10: Impact of Different Chat Metadata and Incorporation Method. *Full* incorporates all three.

*Precision-Recall Tradeoff:* In Figure 2, ToxBuster achieves a consistent precision of 95% across variaous recall levels, indicating its effectiveness in identifying and flagging potentially toxic content. This high average precision demonstrates reliable results for moderation purposes.

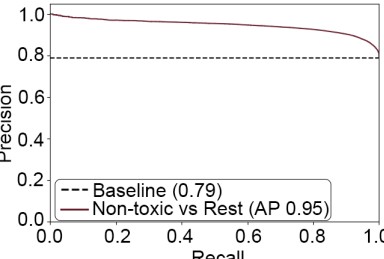

Figure 2: ToxBuster Precision-Recall Curve on R6S. Average precision for non-toxic vs. toxic words is 95%.

*Performance at High Precision Settings:* ToxBuster successfully intercepts 5.38%, 2.07%, and 0.81% of chat lines for R6S when operating at high precision thresholds of 90.0%, 99.0%, and 99.9%, respectively. This emphasizes ToxBuster's potential as a real-time chat moderation solution that incorporates chat history and metadata, which is a capability currently lacking in Perspective API. Table 11 presents the recall rates for each class at high precision levels. At a precision of 99.9%, ToxBuster achieves recall rates ranging from 0.4% to 6.5% across different classes. Notably, the classes of **Spam**, **Scams and Ads** and **Minor Endangerment** demonstrates the highest recall rates, likely due to the distinct language patterns compared to other toxic classes.

*Viability for post-game Moderation:* Table 12 presents a cross-reference of distinct players flagged by ToxBuster, chat-reported players, and reported players (due to disruptive behaviors) over a one-week period in R6S. It is important to note that, at the time of data collection, R6S only al-

| Class Name | 90.0% | 99.0% | 99.9% |
|---|---|---|---|
| Hate and Harassment | **14.22%** | 2.35% | 2.35% |
| Threats | 0.26% | 0.26% | 0.26% |
| Minor Endangerment | 7.32% | **6.69%** | **6.69%** |
| Extremism | 0.50% | 0.50% | 0.50% |
| Scams and Ads | **8.84%** | **4.76%** | **4.76%** |
| Insults and Flaming | 1.18% | 0.39% | 0.39% |
| Spam | **66.63%** | **42.01%** | **6.14%** |
| Other Offensive | 2.09% | 0.72% | 0.72% |

Table 11: ToxBuster Recall Rate per Toxic Class at High Precision Levels for R6S.

lows players to report others for a single reason, meaning players engaging in both toxic chat and disruptive in-game behavior are more likely to be reported for their behavior rather than their chat.

| % of Players | 90.0% | 99.0% | 99.9% |
|---|---|---|---|
| $F/P$ | 29.48% | 11.64% | 7.89% |
| $(F \cap CR)/CR$ | 82.1% | 51.1% | 41.3% |
| $(F \cap \neg CR)/\neg CR$ | 26.44% | 9.36% | 5.96% |
| $(F \cap R)/R$ | 55.49% | 26.92% | 19.57% |
| $(F \cap \neg R)/\neg R$ | 19.18% | 6.37% | 3.86% |

Table 12: Intersection of flagged players (F) by Tox-Buster, chat-reported players (CF) and reported players (R). $CR$ players represent 5.47% of all distinct players and $R$ players represent 25.64%.

We utilize this metric for two estimates. Firstly, we estimate the model's precision based on player reports from the players' perspective, acknowledging that this perspective may slightly differ from actual moderators. Secondly, we estimate the reduction in workload for moderators. By implementing a simple yet effective automated post-game moderation system that focuses on players both flagged by ToxBuster and chat-reported, we can promptly address toxic chat-reported players, leading to a more positive player experience compared to manual inspection by moderators for each instance.

The second row of Table 12 demonstrates that ToxBuster identifies between 41% to 82% of chat-reported players, indicating a high precision from players' perspective. This approach also significantly reduces the workload for moderators when dealing with this set of users. A more sophisticated system could consider the severity level from each toxic class for further improvements.

### 4.7.1 Proactive Moderation

Toxicity in online games and social media platforms often goes unreported, which is a significant issue (ADL, 2022). ToxBuster can help address this problem through proactive light automatic moderation. Table 12 demonstrates that a substantial number of flagged players were not reported by other players. For proactive moderation, we utilize the average number of flagged chat lines per match from chat-reported players as a simple measure. Figure 3 shows that players with more chat reports tend to have a higher number of flagged lines of chat per match. Specifically, for players with more than 3 chat reports, the average number of flagged lines is 5. By applying this criterion, approximately 6.39% of non-chat-reported toxic players would be identified, a start towards addressing this issue.

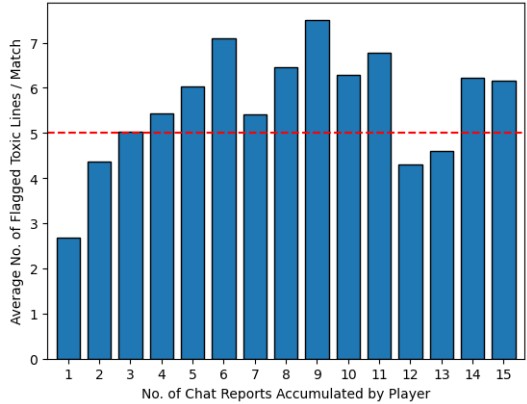

Figure 3: Average number of flagged toxic lines/match over for increasingly chat-reported players. Players are often chat reported when having more than 5 flagged toxic lines.

## 5 Conclusion

In this paper, we propose ToxBuster, a simple and scalable model specifically designed for real-time chat moderation in in-game chat. ToxBuster outperforms baseline models, including Cleanspeak and Perspective API, across multiple datasets and toxic categories. Additionally, ToxBuster can be easily adapted for comment threads of social media platforms. Our study emphasizes the importance of using up-to-date toxic game chat datasets to enhance model robustness, rather than relying on generic toxicity datasets. Furthermore, we demonstrate the seamless transferability between R6S and FH datasets. Ablation studies further validate the significant contributions of chat history and speaker segmentation to ToxBuster's effectiveness.

The precision-recall analysis illustrates Tox-Buster's capability to accurately identify and flag toxic content with high precision and recall. In

an automated post-game moderation scenario, Tox-Buster successfully identifies toxic chat players, flagging over 41% of chat-reported players at a precision level of 99.9%. Additionally, we explore how ToxBuster can partially address the issue of toxic players who are not chat-reported. Overall, ToxBuster presents a robust and efficient solution for automating chat moderation, promoting a safer and more positive gaming experience for players.

## Limitations

Currently, the model's dataset is limited to the English language, with the exceptions of common toxic phrases appearing in in-game chat lines from other languages. Based on results from Perspective API and Jigsaw, we know that the methods presented in this paper can be extended from monolingual to multi-lingual.

ToxBuster will make errors. Only existing patterns of toxicity in the dataset will be detected. Language that closely resemble existing patterns of toxic language could also be incorrectly flagged. As such, the model without any active learning is not suitable for a fully automated moderation. The model also cannot completely replace human moderation.

ToxBuster is intended for in-game chat that will have mentions of in-game events. Hence, phrases that could be considered toxic (a threat) in normal everyday language could be scored as neutral or having less probability of being toxic.

ToxBuster is a step in the right direction towards combatting the many challenges of moderating in-game chat toxicity. In terms of improving toxicity detection, some directions are performing domain adaption on the base language model on unlabeled chat data, continuous learning and adversarial training. The model and dataset can be extended from English to multilingual. Another area is biases and its mitigation. While we have mitigated some during the data annotation phase, we still need to measure biases the model has learned and ways to debias the models without degrading the model performance. Finally, we can also analyze the causes and impacts of toxicity from a player and game design perspective.

## Ethics Statement

As with any language models, ToxBuster will propagate any existing biases in the dataset. We have tried to mitigate biases in the annotation by taking the diversity of the annotators identities into consideration. In our sessions, we recognize that it was hard to recruit those that identify as a woman. We had more success in recruiting those that identified as belonging to marginalized groups (e.g. LGBTQA1+, BIPOC), where half of the annotators self-identifies as belonging to at least one of the marginalized group.

Annotators were also warned about the toxic content they will see. They were given a very lax schedule and allowed to annotate freely at their own pace over a lengthy time period, allowing many breaks if needed.

As stated in the limitations, we are in the process of devising methods to measure bias and debias the model. An adaptation of Kiritchenko and Mohammad (2018) *Equity evaluation corpus* (EEC) will be created to test and measure several categories of social biases such as gender, race, sexual orientation, etc. Meade et al. (2022) also includes a few benchmarks (*Sentence Encoder Association Test* and *Word Embedding Association Test* (May et al., 2019)) and de-biasing methods. De-biasing methods include counterfactual data augmentation (CDA), increasing dropout and projection-based techniques. CDA works by rebalancing the dataset by swapping bias attribute words. As recommended by Blodgett et al. (2020), we have invited and welcomed new researchers from other disciplinary studies, namely from linguistics and psychology.

## Acknowledgements

We wish to thank Ubisoft La Forge, Ubisoft Montreal User Research Lab and Ubisoft Data Office for providing technical support and insightful comments on this work. We also acknowledge funding in support of this work from Ubisoft, the Canadian Institute for Advanced Research (CIFAR AI Chair Program), Natural Sciences and Engineering Research Council of Canada (NSERC) Postgraduate Scholarship-Doctoral (PGS D) Award and Fonds de recherche du Québec - Nature et Technologies (FRQNT) Doctoral Award.

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

## A   Annotator Details

For R6S and FH, annotators were recruited from social media with representation in game experience and self-identification with marginalized groups

taken into consideration. Each annotator had to be at least 18 years old, advanced in English proficiency, reside in the North American time zone and *active* in the respective game. We define *active* as having played the respective game within the last year for at least 16 hours in the player versus player (PVP) mode. After the initial recruitment, a pilot test was conducted to further filter annotators to those that understood the task and aligned themselves on the common definition of toxicity based on examples shown. Each annotator was instructed to highlight the minimum span of contiguous words in a chat line that falls under a toxic category. If a span of words can fall under more than one toxic category, they were to use the most severe category. Each chat line was annotated by three annotators, with full visibility of all previous chat lines of the match available. We also did not show any game related events nor whether the player was reported.

## B   Detailed Toxicity Definitions

Our dataset does not include any audio or visual data, and therefore, categories such as cheating, abuse of play, antisocial actions are not within the scope of this model.

1. **Hate and Harassment:** Identity-based hate or harassment (e.g., racism, sexism, homophobia) or bullying / mobbing (e.g., a group of players bullying one or more players).

2. **Threats:** Threats of violence, physical safety to another player, employee or property, terrorism, or releasing a player's real-world personal information (e.g., doxxing).

3. **Minor Endangerment:** Sexual and/or aggressive actions towards minors or attempts to get minors to perform sexual activities.

4. **Extremism:** Extremist views (e.g., white supremacy), attempts to groom or recruit for an extremist group or repeated sharing of political, religious, or social beliefs.

5. **Scams and Ads:** Fraud / scamming (e.g., including phishing, account stealing, bad trades or theft), posting inappropriate links (e.g., malware, dangerous websites, advertising exploits, etc ) and advertising of websites, services, cheats or rival products.

6. **Insults and Flaming:** Insults or attacks on another player or team (not based on player or team's real or perceived identity)

7. **Spam:** Excessive sharing of the same or similar words, phrases, emojis or sharing (e.g., "kdjfklsjafkldjkla").

8. **Other Offensive Texts:** Any other message not covered in the above categories that is offensive and/or harms a player's reasonable enjoyment of the game.

## C    Model Reproduction Details

Our model is implemented with HuggingFace[4] and PyTorch[5]. Section 3.2.1 and 3.2.2 include explicit information on the model architecture and preprocessing steps. Our model uses default values for BERT. The learning rate is 1e-5, chosen from a hyperparameter search amongst 1e-4, 1e-5 and 1e-6. The learning rate scheduler is set to linear decay with a warmup step ratio of 5%. Training with GeForce RTX 2080 took approximately 7 hours with max train epochs set to 100, early stopping with patience of 5 epochs based on the weighted F1 score.

## D    Sentence-level vs. Token-level

For this experiment, we do not include chat speaker segmentation and use *global* mode for chat history. We analyze in Table 13 the impact of changing the max_token_size for the tokenizer and the level of classification (sentence-level and token-level). Contrary to our beliefs, it would seem that classifying on the sentence level as opposed the token-level is a slightly harder task.

|     | Token | Sentence |
| --- | --- | --- |
| 64  | 77.06 ± 0.88 | 76.12 ± 1.56 |
| 128 | 78.90 ± 0.64 | 79.05 ± 1.39 |
| 256 | 81.41 ± 0.42 | 80.45 ± 1.27 |
| 512 | 82.09 ± 0.39 | 80.88 ± 0.51 |

Table 13: Model mean F1 score on different max token length and classification mode.

## E    Cold-Start Problem

As chat history is used for the model, a cold-start problem may arise. To address this, we examine the results of ToxBuster$_{BASE}$ (is not trained

and doesn't use chat history) and ToxBuster$_{FULL}$ (trained with chat history and speaker segmentation) on the R6S test set. We bin the number of chat lines in the chat history and report the F1-score in Table 14. We can see that using chat history will always help with the performance, although marginally with 0 or 1 lines of chat history and much higher for more lines of chat history. This could indeed explain some of the variance we have reported in the main results. More interestingly, we see a large improvement in performance for 21 - 30 lines and the improvement slightly dropping afterwards, suggesting that this may be the ideal length of chat history for this dataset.

| Chat History | Support | ToxBuster$_{BASE}$ | ToxBuster$_{FULL}$ |
| --- | --- | --- | --- |
| 0 | 96 | 81.69 | 81.72 |
| 1 | 92 | 80.65 | 81.68 |
| 2 - 10 | 811 | 78.15 | 83.43 |
| 11 - 20 | 901 | 77.92 | 82.22 |
| 21 - 30 | 772 | 75.87 | 84.25 |
| 31 - 40 | 717 | 77.35 | 82.75 |
| 41+ | 14,447 | 77.14 | 83.65 |

Table 14: Impact of Context - Model F1 score across differing chat history length.

---

[4] https://huggingface.co/
[5] https://pytorch.org/

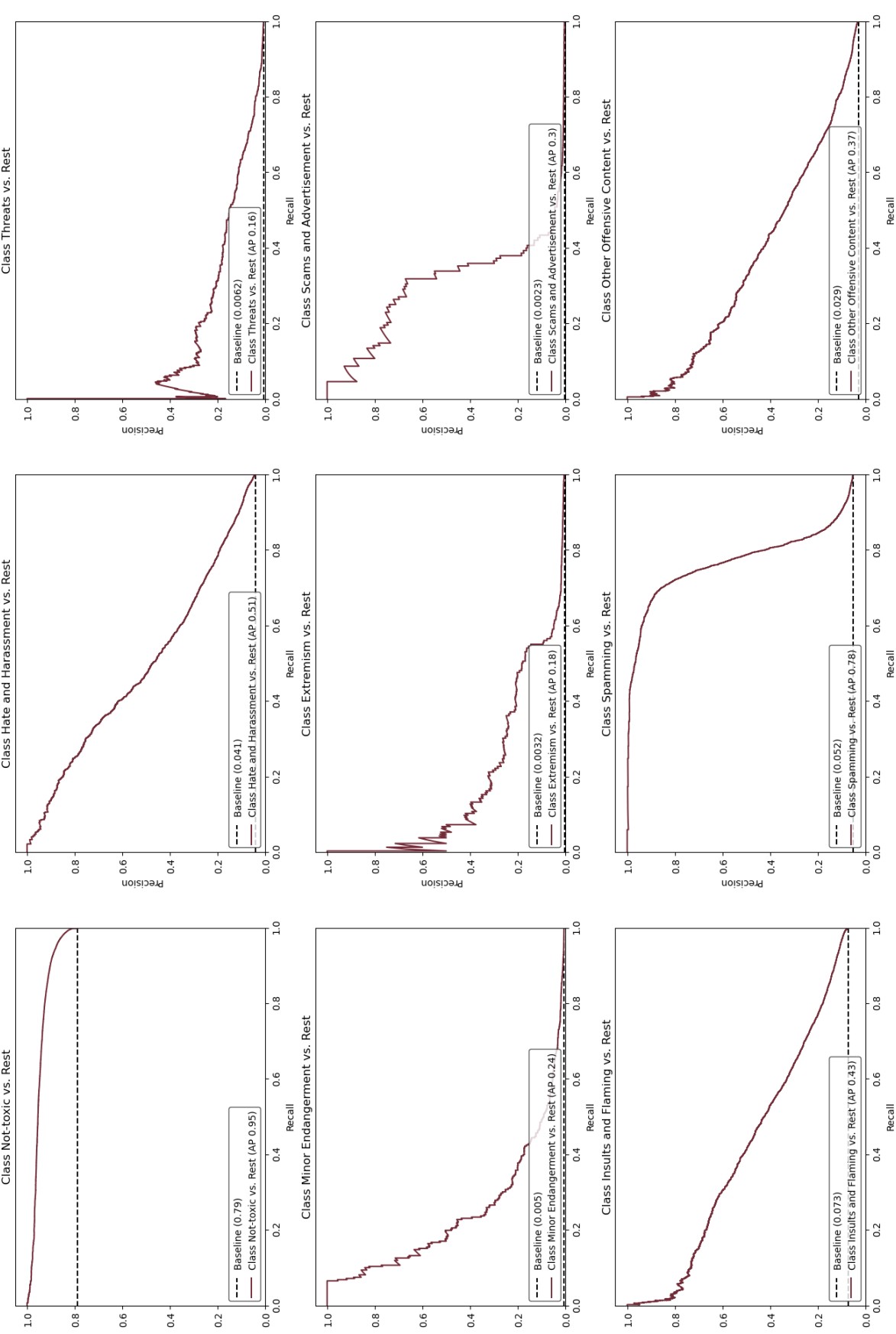

Figure 4: ToxBuster Precision-Recall Curve per Toxic Class on R6S.