# OpenReview forum: "Towards Detecting Contextual Real-Time Toxicity for In-Game Chat"
_EMNLP/2023/Conference — EMNLP 2023 Findings_

### Official Review · Reviewer_yvYH · 2023-08-05

**Typos Grammar Style And Presentation Improvements:** Table 6 should define what metric it …
**Soundness:** 4

**Excitement:**

4: Strong: This paper deepens the understanding of some phenomenon or lowers the barriers to an existing research direction.

**Paper Topic And Main Contributions:**

This paper introduces ToxBuster, a new model for detecting toxicity in game chat. Their model is based on BERT_Base, with two additions. First, they incorporate chat history into the model, testing different "scopes" of history (ie, which chat lines are visible, eg, moderator scope can see all lines and personal can only see lines by the current player). Second, they incorporate chat metadata, including player ID, chat type, and team ID. They test their model on four datasets -- two datasets that they construct (R6S and FH) and two existing datasets (DOTA 2 and CC) -- and compare to three baselines (Cleanspeak, Perspective API, and Detoxify, a model from 2020). They show that their model outperforms the baselines on all datasets, and their full model significantly outperforms BERT_Base on two of the datasets (the ones they curate).

Then, they conduct a series of detailed analyses of ToxBuster. They analyze transferability across datasets and find better transferability within the gaming domain (compared to CC) and within similar time periods. They find that ToxBuster, when trained on R6S and adapted to FH, performs very well. They also conduct class-wise evaluations, showing differential performance across different types of toxicity; an error analysis of false positives and negatives; ablation studies; evaluations of recall in high-precision settings; and comparisons to ToxBuster's high-precision predictions to actually flagged players, finding that ToxBuster flags a large proportion of these users.

**Questions For The Authors:**

A. It seems like this is not the first model to incorporate context into toxicity or hate speech detection (as mentioned in Related Work). Have you tried implementing other models that use context and comparing against them? Or, as a weaker baseline, using the APIs on each line but using the score from the lines in the history to help predict whether the current one is toxic?
B. Are the ablation analyses in Section 4.6 only computed on FH and R6S? Do you know why these additions (chat history and metadata) don't seem to improve performance as much for the other two datasets in Table 5?

**Reasons To Accept:**

- ToxBuster outperforms the baseline models across four datasets
- Strong results from transferability and adaptability results
- Thorough evaluation: class-wise, error analysis
- Promising results in post-game moderation, with a high proportion of chat-reported players flagged by ToxBuster

**Reasons To Reject:**

- Method additions beyond BERT_base are relatively simple (chat history and chat metadata); adding context is commonly done
- These method additions also don't improve performance much for two of the datasets, comparing ToxBuster-full and ToxBuster-base
- Performance varies substantially across toxicity classes and is much worse for certain classes, eg, F1 below 30

**Reproducibility:**

4: Could mostly reproduce the results, but there may be some variation because of sample variance or minor variations in their interpretation of the protocol or method.

**Reviewer Confidence:**

4: Quite sure. I tried to check the important points carefully. It's unlikely, though conceivable, that I missed something that should affect my ratings.

---

> ### Author Rebuttal · Authors · 2023-08-28
>
> # R1: Simplistic Method Additions; adding context is commonly done
>
> We acknowledge that contextual toxicity detection has been explored in the literature, referring back to our related works section. However, our approach **distinguishes itself by emphasizing the conversational aspect**, which has not been extensively addressed before. Specifically, our model incorporates the entire chat history, offering varying scopes of history (e.g., moderator scope vs. player scope) and chat metadata. While these additions may seem straightforward, they represent a novel contribution to the field of toxicity detection.
>
>
> # R2:  Low Improvement on CC and DOTA 2 Dataset
>
> We appreciate your observation regarding the performance discrepancy between ToxBuster_BASE and ToxBuster_FULL on certain datasets. Regarding the CC dataset, it primarily consists of **comments, which tend to be longer in nature (~55 words per line)**. In such cases, the marginal gains from context are expected, as evidenced by similar studies focusing on tweets and comments. In contrast, in-game chat, as seen in R6S and FH, is notably shorter, where the inclusion of all possible chat history is particularly effective for toxicity detection. As for the DOTA 2 dataset, it was adapted from a different task with two levels of annotation. The first level of annotation for toxicity is based off of a list of keywords, making the contextual information less critical. The second level of annotation on the sentence level is human-annotated for explicit and implicit toxicity, where explicit toxicity had to contain words that were labeled toxic. Only about 7% of the lines had implicit toxicity and could not be labeled as toxic with a keyword-based annotation. We provide the F1-score of ToxBuster_BASE and ToxBuster_FULL in the table below where we group the samples by the original class of the human-annotation. We see that there is a **higher performance gain in the implicit class** compared to the explicit class, as implicit toxicity benefits more from context as opposed to the explicit class.
>
> | Original Class | Support | ToxBuster_BASE | ToxBuster_FULL |
> |---------|------------|-------------------------|------------------------|
> | Explicit  | 1950  | 77.81   | 78.58  |
> | Implicit  | 146    | 64.27   | 67.54  |
> | Non-Toxic | 2123 | 91.14 | 91.16 |
>
> # R3: Varied Performance; Low F1-Score on Certain Categories
>
> We recognize the variability in performance across toxicity classes, particularly for categories with limited training data such as threats and minor endangerment. We acknowledge it in the class-wise evaluation section and plan to **address it in future work**. Our ongoing research focuses on methods to improve the F1 scores for each category. Furthermore, we intend to **explicitly mention the low-performing categories in the limitations section** in the camera-ready version..
>
> # Q1: Baseline with other contextual toxicity detection models.
>
> We appreciate your suggestion regarding the comparison with other contextual models and the weak baseline. To clarify, while previous contextual toxicity detection models have explored aspects like metadata (e.g. article title, usernames), our emphasis is on conversational metadata, which presents a unique context not addressed by existing models. For the preceding comment, we **point towards Table 13 in the Appendix**. While not directly comparable to only including the preceding comment, the lower the max token length, the less chat history we have. We see that as we increase the max token length, i.e. the more chat history that is available, the better our model performs on the R6S and FH dataset. We would be happy to include results of limiting our model to a specific number of lines of chat as part of the Appendix for the camera-ready version.
>
> As for the use of APIs on each line and incorporating historical scores, it is an intriguing idea. Designing this baseline was not straightforward for us, but based on your suggestion, we’re thinking of training a classifier on two features, the average of toxicity scores of the past chat line and the current toxicity score of the chat line from Perspective API to predict the toxicity label of the current chat line. Due to the short time of the rebuttal, we will be **adding this as a baseline in the camera-ready version**. Although we want to highlight that at the end it is not ideal for the solution to depend on a third-party API, and a solution like ToxBuster which can be deployed in house is more practical.
>
> # Q2:  Clarification on Ablation Analysis
>
> Indeed, the ablation analyses were **conducted solely on the FH and R6S datasets, which are the best match with our task given being from in-game chats and being properly labeled**. As for the performance variation across the CC and DOTA 2 datasets which are not ideal for our task, CC is not from the game domain, and DOTA 2 is not properly labeled as we discussed earlier in R2. We believe these factors contribute to the observed performance differences and will provide further clarification in the revised manuscript.
>
>
>
> Once again, we sincerely thank you for your valuable feedback and will diligently address your comments and suggestions in the camera-ready version.

---

### Official Review · Reviewer_BWUQ · 2023-08-05

**Soundness:** 3

**Excitement:**

3: Ambivalent: It has merits (e.g., it reports state-of-the-art results, the idea is nice), but there are key weaknesses (e.g., it describes incremental work), and it can significantly benefit from another round of revision. However, I won't object to accepting it if my co-reviewers champion it.

**Paper Topic And Main Contributions:**

This paper addresses the problem of real-time toxicity detection for in-game chat. They propose ToxBuster, a BERT-based model that employs chat history and speaker segmentation features for detection. Experiments show that the proposed model outperforms three off-the-shelf general-purpose toxicity detection models. They presented ablation studies to show the effectiveness of each sub-module. Cross-prediction experiments across different datasets suggest that it can achieve a reasonable performance for predicting toxicity for different games. This paper could make a resource contribution to the NLP researchers on abusive languages. There could be a possibility of method-wise contribution, but it seems to require further investigation.

**Reasons To Accept:**

A1. This paper tackles an important problem of toxic speech detection in online platforms. The new annotation dataset could be a valuable resource to the NLP researchers on abusive languages because they were constructed on real-time chat on online games.
- A downside regarding this point is that the whole paper is framed for other types of contributions, which are not adequately supported by the paper and experiments.

A2. They conducted various experiments to understand the detection ability of the proposed method. In particular, the cross-dataset experiment provided insights into practical considerations on using the ML model. It was interesting to see the CC-based model can achieve a reliable performance on the game chat data.

**Reasons To Reject:**

R1. The paper does not provide enough details to reproduce the proposed method.
- It misses the details of how the chat history feature is implemented for the CC dataset.
- I don’t clearly understand how the chat speaker segmentation embeddings are represented and incorporated with the BERT input embeddings.
- Some datasets, even a game dataset, may not provide all the information required for building the model. For example, there could be a game that does not provide Team ID for chat data collection. More ablation experiments could be conducted to understand the effects of each component of chat segmentation features.

R2. While the experiments show the importance of considering chat history for better detection, the high variance in the evaluation of chat segmentation features suggests that its attribution to performance is not clearly understood.
- History features may suffer from cold-start problems, which could be analyzed in additional experiments.

W3. In the Introduction, this paper argued that “most existing approaches either neglect context or yield only marginal improvements and not fit for real-time moderation for an in-game chat” to support its method-wise contribution. I believe some of the arguments are wrong, and others require more justification or experiments.
- The idea of considering additional contexts has been widely discussed in the context of abusive language detection (https://arxiv.org/abs/1808.10245, https://aclanthology.org/W19-3508/).
- Despite some analyses presented in the paper, I was not fully convinced of the real-time detection capability of the proposed method. A potential issue is a cold-start problem, as I describe above.

**Reproducibility:**

3: Could reproduce the results with some difficulty. The settings of parameters are underspecified or subjectively determined; the training/evaluation data are not widely available.

**Reviewer Confidence:**

3: Pretty sure, but there's a chance I missed something. Although I have a good feel for this area in general, I did not carefully check the paper's details, e.g., the math, experimental design, or novelty.

---

> ### Author Rebuttal · Authors · 2023-08-28
>
> # A1: Framing of Contributions
>
> We appreciate the reviewer's recognition of the importance of our paper's contributions to toxicity detection in online game chat. We'd like to clarify that our primary **contributions are twofold**:
>
> 1. **Innovative Model**: We introduce ToxBuster, a BERT-based model that leverages chat history and speaker segmentation features for toxicity detection. Previous contextual toxicity detection models either only consider the preceding comment rather than the full history or use metadata such as article titles, usernames or game-specific slang (CONDA: a CONtextual Dual-Annotated dataset for in-game toxicity understanding and detection). With both chat history and speaker segmentation, our model ***addresses the conversational aspect for in-game chat that is novel in the context of toxicity detection***.
> 2. **Practical Insights**: We provide valuable insights into the real-world applicability of our model, including cross-dataset experiments, post-game moderation analysis, and proactive moderation potential. First, the ***transferability between games*** and the CC dataset shows how our model and dataset is useful for game companies with more than one game or inviting collaborations between game companies. Hence, we also use categories defined under the Fair Play Alliance (FPA). Second, we provide post-game moderation analysis as the closest baseline to estimate the ***effectiveness of the model in a real-world setting***. Finally, we also provide the proactive moderation aspect as another potential use for real-world settings. While exploratory, it showcases how this ***model can proactively protect players in real-time***, i.e. muting a player for excessive toxic usage while the game is still active. These experiments showcase the model's practicality in a real-world setting.
>
> We believe that these contributions are well-supported by our experiments and analysis.
>
>
> # R1: Method Reproducibility
>
> **1. Chat History for the CC dataset**:
>
> We only briefly state that the comment history is created by “tracing all parent comments”.  To further clarify this, we **provide the following explanation**:
> > In the original Jigsaw’s Civil Comments dataset, each comment is accompanied with metadata such as, “id”,  “parent_id” and  “article_id”. In this case, a document is one full comment thread within the same article (i.e. identical “article_id”), starting from the root comment all the way down its child comments. Using the “parent_id”, we can build a tree of comments by backtracking the parent-child relationship formed when linking comments where the “parent_id” of a comment matches the “id” of another. We note that an article can have many trees of comment threads and that each path from the root of one tree to its leaf would then be one document. For the train & test split, we split by articles to prevent test leakage. The processed CC dataset and the corresponding processing code are also included in the supplementary material, ensuring complete transparency and ease of replication.
>
> **2. Chat Speaker Segmentation Representation & Implementation**:
>
> For the chat speaker segmentation, we would like to **update Table 4 to increase the clarity**.
>
>  | # | Line                         | PlayerID | Chat Type | TeamID |
>  |---|------------------------------|----------|-----------|--------|
>  | 1 | (Team) Apple: Hf          | 6        | Team      | 1      |
>  | 2 | (All) Banana: Hf           | 5        | All       | 1      |
>  | 3 | (Team) Grape: Which site? | 1        | Team      | 0      |
>  | 4 | (Team) Orange: A           | 0        | Team      | 0      |
>  | 5 | (All) Orange: Glhf         | 0        | All       | 0      |
>
> We would also be **updating the paragraph about the metadata to the following**:
>
> > For the three chat metadata, *playerID* and *teamID* dynamically changes based on the player of the current chat line. *ChatType* can be either **team** or **all**, indicating whether a line is exclusive to the **team** or broadcasted to **all** players. *PlayerID* is the unique identifier for the player associated with the chat line, starting from **0** and bounded by the number of teams times the team size. For consistency, the player of the current chat line is always **0**. For other players on the same team, the identifier is incremented based on the recency of that player’s chat line. For players on the other team, the identifier starts from the size of the team, e.g, in a 5 v 5 game, the most recent opponent that has typed in chat will be **5**. With this scheme, the *playerID* can be extended to even Battle Royal games, where there can be multiple enemy teams. *TeamID* is the unique identifier of the team the current player belongs to. For consistency, the current player is always team **0**. The enemy team would be team **1**. For battle royal games, this scheme can be extended similarly to the playerID. The last three columns in Table 4 describe the *playerID*, *chatType* and *teamID* when detecting toxicity for line 5.
>
> Finally, we will be **adding a new paragraph about the model architecture**, as it is only briefly mentioned in the Figure 1 captions but not in text to the following:
>
> > As shown in Figure 1, ToxBuster is a BERT-based model with an additional three inputs provided. BERT’s input embedding is the sum of the position and token encoding. For ToxBuster, we introduce an encoding for each of the TeamID, Chat Type and PlayerID that corresponds to the token. Similar to BERT, the input embedding would then be the sum of all 5 encodings: Token, Position, TeamID, Chat Type and PlayerID.
>
> **3. Missing game-specific metadata:**
>
> We acknowledge that not all datasets may contain game-specific metadata, such as the Chat Type or TeamID. To address this issue, we **refer to Table 10**, the ablation study for the chat speaker segmentation which provides the impact of each new input encoding (Player ID, Chat Type, Team ID) individually and with all included. If the reviewer is asking for a comprehensive ablation study with all possible combinations, we can include that as a table in the Appendix in the camera-ready version.
>
> # R2: Cold-Start Problem
>
> We appreciate the reviewer for bringing up the concern of cold-start as our model is trained with chat history and may be dependent on it, causing a cold-start problem. We address this by examining the results of ToxBuster_BASE (is not trained and doesn't use chat history) and ToxBuster_FULL (trained with chat history and speaker segmentation) on the R6S test set. We bin the number of chat lines in the chat history and report the F1-score. We can see that using **chat history will always help with the performance**, although **marginally with 0 or 1 lines of chat history** and much **higher for more lines of chat history**. This could indeed explain some of the variance we have reported in the paper. More interestingly, we see a large improvement in performance for 21 - 30 lines and the improvement slightly dropping afterwards, suggesting that this may be the ideal length of chat history for this dataset. This is a useful insight that can be further explored in future works to improve the model performance and we thank the reviewer again for the comment which led to the new analysis.
>
> | # of Lines in Chat History | Support | ToxBuster_BASE | ToxBuster_FULL |
> |----------------------------|---------|----------------|----------------|
> | 0                          | 96      | 81.69          | 81.72          |
> | 1                          | 93      | 80.65          | 81.68          |
> | 2 - 10                     | 811     | 78.15          | 83.43          |
> | 11 - 20                    | 901     | 77.92          | 82.22          |
> | 21 - 30                    | 772     | 75.87          | 84.25          |
> | 31 - 40                    | 717     | 77.35          | 82.75          |
> | 41+                        | 14447   | 77.14          | 83.65          |
>
>
>
>
> # W3: Contextual Toxicity Detection Models
>
> We appreciate the reviewer's feedback, especially with the citation of two papers on contextual toxicity detection models. We acknowledge that incorporating context for toxic or abusive language detection is not new, as stated in our related works. We note that many papers focus on tweets or comment threads, which have much longer words per line compared to in-game chat. We would also like to stress that to our knowledge, **none of the toxicity models consider more than the preceding comment or line**. They also **do not consider the conversational aspect** when there is more than one preceding comment. Regarding the comparative studies the reviewers cited, they actually **support our claim** further.:
>
> * **Comparative Studies of Detecting Abusive Language on Twitter**: The paper does a comprehensive study on different models while including “context tweets”. Based on our understanding and the examples the authors provide, this only includes the preceding tweet, i.e. the tweet that is replied to. This study also **confirms our statement** of “yielding only marginal improvements” as shown when the overall F1 score of their CNN and RNN models actually decrease when context is provided. CNN with context does perform better on the HATE class with an increase of  .05 for its F1 score.
>
> * **Pay “Attention” to Your Context when Classifying Abusive Language**:  One of the main results of the paper is the usage of “context attention” which is jointly learned when training the model. In their results of comparing whether to stack Bi-LSTM and whether to use self-attention or context-attention, the stacked Bi-LSTM with context attention prevails with an increase of the F1-score by less than 1 in 3 of their datasets. While interesting, we note that 1) it only changes **how it attends to the context around the word in the given sentence** (model architecture), 2) does not take into account any conversational aspect.
>
>
> # W3:  Real-time Detection Capability
> We understand the reviewer's concern regarding real-time detection capability. We show **real-world applicability of our model by including cross-dataset experiments, post-game moderation analysis, and proactive moderation potential**. As all previous chat history can be provided, a contextual real-time in-game chat toxicity detection is feasible. We thank the reviewer for bringing the concern of the cold-start. The results shown in R2 should help answer that concern. Additionally, we would like to highlight that this research is in **collaboration with a company and is undergoing the industrialization process, i.e.  real-world deployment**.
>
> We sincerely appreciate the reviewer's constructive feedback, and we are committed to addressing these concerns in the revised manuscript. We believe these updates will strengthen our paper and make our contributions more evident and reproducible. Thank you for your valuable input.

---

### Official Review · Reviewer_KywN · 2023-08-05

**Soundness:** 3

**Excitement:**

3: Ambivalent: It has merits (e.g., it reports state-of-the-art results, the idea is nice), but there are key weaknesses (e.g., it describes incremental work), and it can significantly benefit from another round of revision. However, I won't object to accepting it if my co-reviewers champion it.

**Paper Topic And Main Contributions:**

The authors study the problem of detecting toxicity on gamming platforms. They introdyce a model called ToxBuster which leverages chat history and meta-data for toxicity detection. They further experiment with ToxBuster on several gamming platforms.

**Reasons To Accept:**

1.	The authors study an interesting an important problem- toxicity detection.
2.	There are several experiments like performance comparisons, error analysis, ablation studies, and some exploratory experiments

**Reasons To Reject:**

1.	The novelty of the work seems limited. It is not clear as to why detecting toxicity in gaming platforms unique compared to other settings for toxic content detection.
2.	The proposed model is trivial. The authors propose to use BERT which crunches on chat history and current chat. There are no innovations on the architecture which are different from existing toxicity detection works.
3.	I would recommend the authors to clarify their innovations in their work.
4.	Furthermore, since the model simply feeds on a concatenation of chat history, it would be interesting to see how LLMs do in such cases (if possible to check).

**Reproducibility:**

3: Could reproduce the results with some difficulty. The settings of parameters are underspecified or subjectively determined; the training/evaluation data are not widely available.

**Reviewer Confidence:**

5: Positive that my evaluation is correct. I read the paper very carefully and I am very familiar with related work.

---

> ### Author Rebuttal · Authors · 2023-08-28
>
> # R1: The Uniqueness of Toxicity Detection in Gaming Platforms
>
> We understand your concern about the uniqueness of our work in comparison to other toxicity detection settings. To clarify, toxicity detection in gaming platforms presents distinct challenges that differentiates it from other settings. In gaming platforms, in-game chat messages are **exceptionally brief, often containing less than five words, contains abundant usage of slang, and are susceptible to frequent misspellings.** This starkly contrasts with platforms such as social media or news article comments, where text is typically much longer such as the CC dataset that  averages around 55 words per line. Moreover, our model incorporates speaker segmentation, a crucial component to capture the essence of in-game chat conversations, as these often involve interactions between multiple users and different teams. This distinct **focus on the conversational aspect** is a departure from existing approaches that predominantly concentrate on single-line comments. We will further emphasize these unique aspects in our paper to provide a more comprehensive understanding of the nuances of toxicity detection for in-game chat.
>
> # R2 & R3: Innovations in Architecture
>
> We appreciate your feedback on the novelty of our architecture.  To address this concern, we'd like to underscore that our approach introduces several innovative elements that distinguish it from existing toxicity detection methods.
>
> Specifically, we **incorporate more than just the previous comment** (Toxicity Detection: Does Context Really Matter?) by utilizing all preceding chat history available as well. Additionally, we integrate speaker metadata, including TeamID, Chat Type, and PlayerID, as integral parts of the model. While previous research on toxicity detection has explored metadata aspects like article titles, usernames or game-specific slang (CONDA: a CONtextual Dual-Annotated dataset for in-game toxicity understanding and detection), none have **addressed the conversational aspect with this level of granularity**, especially in contexts where more than just the preceding line of text is used.  Finally, we also **modify the BERT model to incorporate the chat speaker segmentation as an additional input**. For better clarity, we will include the following detailed explanation of how we modify the BERT model in the paper.
>
> > As shown in Figure 1, ToxBuster is a BERT-based model with an additional three inputs provided. BERT’s input embedding is the sum of the position and token encoding. For ToxBuster, we introduce an encoding for each of the TeamID, Chat Type and PlayerID that corresponds to the token. Similar to BERT, the input embedding would then be the sum of all 5 encodings: Token, Position, TeamID, Chat Type and PlayerID.
>
> # R4. Model Simplicity and Comparison with LLMs
>
> We acknowledge the reviewer's interest in comparing our model with LLMs and how they perform in such cases. First, we want to highlight that our model goes through specific data preprocessing before concatenating chat history. In the paper, we explore different filters for the chat history, focusing on personal, team, global, and moderator filters, determining their relevance and model performance impact to the current chat line. Our experimentation indicates that the global filter, which simulates the chat experience with all chat enabled, yields the most effective results. As previously discussed in R2 & R3, we also incorporate speaker segmentation to capture the conversational nuances present in gaming platform interactions.
>
> Furthermore, we'd like to highlight that deploying LLMs or calling GPT-3.5 / GPT-4 with OpenAI API for discriminative tasks such as toxicity detection is cost-prohibitive, as supported by the findings in "Open, Closed, or Small Language Models for Text Classification?" (https://arxiv.org/abs/2308.10092). The scalability and speed advantages of BERT-based models, i.e. ToxBuster, and its cost-effectiveness are particularly crucial for the high volumes of game chat typical of multiplayer games. We note that Perspective API, one of the baseline we compare to, is also a BERT-based model that is capable of real-time scenarios (response time ~100ms).

---

### Meta-Review · Area_Chair_yC1u · 2023-09-15

**Recommendation:** 2

**Metareview:**

The authors

However, the proposed analysis shows some drawbacks and limitations:
1) The novelty of the work seems limited. It is not clear why detecting toxicity in gaming platforms is unique compared to other settings for toxic content detection;
2) The chat speaker segmentation embeddings should be better described, as well as discussing how it is integrated into the architecture.
3) The proposed approach slightly improve the performance w.r.t. the other approaches.
4) Performance varies substantially across toxicity classes and is much worse for certain classes, eg, F1 below 30

---

### Decision · Program_Chairs · 2023-10-07

**Decision:**

Accept-Findings

**Comment:**

The authors

However, the proposed analysis shows some drawbacks and limitations:
1) The novelty of the work seems limited. It is not clear why detecting toxicity in gaming platforms is unique compared to other settings for toxic content detection;
2) The chat speaker segmentation embeddings should be better described, as well as discussing how it is integrated into the architecture.
3) The proposed approach slightly improve the performance w.r.t. the other approaches.
4) Performance varies substantially across toxicity classes and is much worse for certain classes, eg, F1 below 30